# Differences in Medium-Induced Conformational Plasticity Presumably Underlie Different Cytotoxic Activity of Ricin and Viscumin

**DOI:** 10.3390/biom12020295

**Published:** 2022-02-11

**Authors:** Pavel Volynsky, Diana Maltseva, Valentin Tabakmakher, Eduard V. Bocharov, Maria Raygorodskaya, Galina Zakharova, Elena Britikova, Alexander Tonevitsky, Roman Efremov

**Affiliations:** 1Shemyakin-Ovchinnikov Institute of Bioorganic Chemistry RAS, Miklukho-Maklaya Str. 16/10, 117997 Moscow, Russia; tabval@yandex.ru (V.T.); edvbon@mail.ru (E.V.B.); tonevitsky@mail.ru (A.T.); r-efremov@yandex.ru (R.E.); 2International Laboratory of Microphysiological Systems, Faculty of Biology and Biotechnology, National Research University Higher School of Economics, Myasnitskaya ul. 20, 101000 Moscow, Russia; dmaltseva@gmail.com; 3School of Biomedicine, FEFU Campus, Far Eastern Federal University, 10 Ajax Bay, Russky Island, 690922 Vladivostok, Russia; 4Moscow Institute of Physics and Technology, Institutskiy per., 9, 141701 Dolgoprudny, Russia; 5Scientific Research Center Bioclinicum, Ugreshskaya Str. 2/85, 115088 Moscow, Russia; maria.raygorodskaya@gmail.com (M.R.); galina.s.zakharova@gmail.com (G.Z.); 6Institute of Bioorganic Chemistry NASB, Kuprevich St. 5, 220141 Minsk, Belarus; britikova@iboch.by

**Keywords:** membrane trafficking of proteins, medium effects on protein structure, molecular dynamics, type II ribosome inactivating proteins, protein–membrane interactions

## Abstract

Structurally similar catalytic subunits A of ricin (RTA) and viscumin (MLA) exhibit cytotoxic activity through ribosome inactivation. Ricin is more cytotoxic than viscumin, although the molecular mechanisms behind this difference are still poorly understood. To shed more light on this problem, we used a combined biochemical/molecular modeling approach to assess possible relationships between the activity of toxins and their structural/dynamic properties. Based on bioassay measurements, it was suggested that the differences in activity are associated with the ability of RTA and MLA to undergo structural/hydrophobic rearrangements during trafficking through the endoplasmic reticulum (ER) membrane. Molecular dynamics simulations and surface hydrophobicity mapping of both proteins in different media showed that RTA rearranges its structure in a membrane-like environment much more efficiently than MLA. Their refolded states also drastically differ in terms of hydrophobic organization. We assume that the higher conformational plasticity of RTA is favorable for the ER-mediated translocation pathway, which leads to a higher rate of toxin penetration into the cytoplasm.

## 1. Introduction

AB-type protein toxins containing a catalytically active A subunit and a cell-binding B subunit are often used to investigate various stages of intracellular transport [1,2]. Ricin is the most extensively studied AB-toxin, and its spatial structure is well described [3,4,5]. It represents a heterodimer consisting of two subunits linked by a disulfide bond. The catalytic A subunit of ricin (RTA) irreversibly inhibits ribosomes causing cell death [2]. RTA specifically and irreversibly hydrolyses the N-glycosidic bond of the adenine residue at position 4324 (A4324) within the 28S rRNA, but leaves the phosphodiester backbone of the RNA intact [6]. The ricin targets A4324 that is contained in a highly conserved sequence of 12 nucleotides universally found in eukaryotic ribosomes. This sequence, termed the sarcin–ricin loop, is important in binding elongation factors during protein synthesis [7]. The depurination event rapidly and completely inactivates the ribosome, resulting in toxicity from inhibited protein synthesis.

After binding to the cell surface, ricin is endocytosed and transported in a retrograde manner to the endoplasmic reticulum (ER) before being translocated to the cytosol [8]. By this moment, the disulfide bond is reduced, and RTA is separated from the holotoxin. This is accompanied by changes in its conformation: a relatively hydrophobic patch that had been previously occluded by the cell-binding subunit (RTB) is exposed, and the mobility of the terminal fragments is increased [9]. The intracellular transport of ricin is not fully understood.

RTA utilizes the ER-associated degradation (ERAD) quality control system to retrotranslocate across the ER membrane [10]. The proteins are unfolded during retrotranslocation. In order to regain its catalytic activity after translocation to the cytosol, the unfolded catalytic subunit needs to be properly refolded. It was shown that the fate of RTA after translocation to the cytoplasm directly depends on the ability of the protein molecule to refold [11]. The mutant RTA form, which is unable to fold properly immediately after translocation, undergoes ubiquitination and proteolytic degradation, whereas native RTA after translocation avoids both.

In order to understand the translocation process, several mutant RTA variants [12,13], pharmacological agents disturbing the intracellular transport [14,15], and point inhibition of target proteins have been applied [8,14,16]. It has been shown that delivery of competent ricin molecules to ribosomes can be assisted by diverse interactions with the ER membrane components, emphasizing the importance of understanding its structural/dynamic properties not limited to those defining its catalytic activity.

Recently we have described in detail the large-scale conformational changes of the catalytic A subunit of viscumin from the plant *Viscum album* (MLA) [17]. Ricin and viscumin are structural homologs possessing cytotoxic activity; both are classified as type II ribosome-inactivating proteins (RIP) [18,19]. However, ricin is considered as a dangerous substance, while viscumin is used as a pharmaceutical agent, mainly due to its lower cytotoxicity in comparison with ricin [18]. The reasons for this are not fully understood, but it is often explained by different carbohydrate specificities of their B subunits [20]. However, the ribosome inactivation capabilities of the catalytic A subunits should also be compared. Alternatively, higher ricin toxicity may be due to the differences in the membrane-induced conformational rearrangements of the catalytic A subunits. This can lead to different rates of translocation of the toxins into the cytoplasm. Unfortunately, analysis of such transient protein states in a living cell is practically impossible without the toxin modifications that can cause structural distortions and difficulties in interpretation of the results. Therefore, molecular modeling appears to be a viable alternative that potentially allows understanding the subtle differences between the mechanisms of delivery of the two toxins apparently sharing a common toxicity mechanism while having substantially different toxicity levels.

In this study, we address the aforementioned issues using a combination of biochemical and computational methods. First, we measured the cytotoxicity of both RIPs against HT29 cells and evaluated the factors that can affect the observed difference in the toxins’ action. Because the latter critically depends on the ability of both proteins to interact with the cell membrane, the number of binding sites for RIPs RTA and MLA on the HT29 plasma membranes was measured using In-Cell ELISA analysis. In addition, the proportions of ribosomes inactivated by the toxins were evaluated with the help of qPCR. Based on these data, it was suggested that the observed differences in cytotoxicity are associated with the ability of RTA and MLA to undergo structural/hydrophobic rearrangements when trafficking through the ER membrane. This hypothesis was further checked in computational experiments. The following questions were addressed in silico: (1)How plastic are the structures of both proteins—how do they react to changes in the properties of the medium? The term plasticity in this case refers to the degree and speed of the structural rearrangements caused by the effects of the environment. To solve these problems, we performed atomistic molecular dynamics (MD) simulations of proteins in media of different polarity—in water, 8M urea (chosen as the standard denaturing solvent), and a chloroform/methanol mixture, which mimics the water/membrane interface.(2)What are the features of the structural and hydrophobic organization of partially and/or fully refolded states of proteins obtained via MD? To answer this question, we employed the original methods of in silico protein mapping and identification of the putative membrane binding sites. The latter were assessed through Monte Carlo (MC) simulations of RTA and MLA in the presence of the implicit membrane.

## 2. Materials and Methods

### 2.1. Reagents

The following reagents were used for cell culturing: McCoy’s cell culture medium (Gibco, Paisley, Scotland); fetal bovine serum (FBS; Gibco, Brazil); 100× PenStrep antibiotic mixture (Gibco, Grand Island, NE, USA); 1× Dulbecco’s phosphate buffer (DPBS; Gibco, Paisley, Scotland); 0.25% trypsin–EDTA solution with Hanks’ salts (PanEko, Moscow, Russia); 3-(4,5-dimethylthiazol-2-yl)-5-(3-carboxymethoxyphenyl)-2-(4-sulfophenyl)-2H-tetrazolium, inner salt (MTS; Promega, Madison, WI, USA), and an electron coupling reagent phenazine methosulfate (PMS; Promega, Madison, WI, USA).

Molecular biology experiments were performed using the following reagents: 10× phosphate-buffered saline (PBS), pH 7.4 (Gibco, New York, NY, USA); Tween-20 (Panreac, Spain); β-lactose (Sigma, St. Louis, MO, USA); bovine albumin (PanEko, Gorky Leninskie, Russia); Qiazol Lysis Buffer (Qiagen, MD, USA); mRNA isolation kit miRNeasy micro kit (Qiagen, Hilden, Germany); SuperScript VILO cDNA Synthesis Kit (Invitrogen, Calsbad, USA); qPCR Mix HS Sybr (Evrogen, Moscow, Russia); conjugate of streptavidin with horse reddish peroxidase Streptavidin HRP (Invitrogen, Frederick, MD, USA); 3,3’,5,5’-tetramethylbenzidine solution (TMB; Life Technologies, Frederick, MD, USA); biotinamidohexanoic acid N-hydroxysuccinimide ester (Sigma, St. Louis, MO, USA), all other reagents were from Sigma.

### 2.2. Proteins and Conjugates Preparation

Ricin and viscumin were purified from *Ricinus communis* seeds and green parts of *Viscum album*, as described previously [21]. Conjugates of ricin and viscumin with biotin were prepared using biotinamidohexanoic acid N-hydroxysuccinimide ester (biotin-NSE) by a modification of the methods recommended by the manufacturer. The ratio of covalently-bound RIP and biotin was about 1:3. The quality of native and biotinylated toxins was controlled by gel electrophoresis, ELISA, ELLA, and cytotoxicity assays.

### 2.3. Cell Culturing and Cytotoxic Effect of Ricin and Viscumin

The routine culturing of the human colorectal adenocarcinoma HT-29 cells was performed in McCoy’s medium with 10% FCS, as described previously [22]. The measurements of the cytotoxic effect of ricin and viscumin on the HT-29 cell line were carried out as described in [23], with some modifications. Ricin or viscumin was added to cells for 1 h in one of the following concentrations: 0, 1 × 10^−10^ M, 1 × 10^−9^ M, 1 × 10^−8^, and 1 × 10^−7^ M (which was equivalent to 0.0066, 0.066, 0.66, and 6.6 µg/mL for ricin and 0.0065, 0.065, 0.65, and 6.5 µg/mL for viscumin), and then placed in a medium without RIP and incubated for 5 days. Cell viability was assessed using MTS reagent. Each concentration of lectins was had five replicates. The data presented are the results of three independent experiments. Statistical analysis of the survival curves was performed using the program RStudio and the drc software package extension [24].

### 2.4. In-Cell ELISA Analysis of Ricin and Viscumin Binding on the Surface of HT-29 Cells

The HT-29 cells were seeded at 3 × 10^4^ cells per well in 96-well (Corning) and incubated at 37 °C and 5% CO_2_ until confluency. Then cells were washed and fixed by 15 min incubation with a mixture of 1% paraformaldehyde and 0.1% glutaraldehyde in 1× PBS. After that the wells were washed, and the non-specific binding sites were blocked by overnight incubation with 0.1% BSA solution in 1× PBS containing 0.05% Tween 20 (PBST). The biotinylated ricin or viscumin 650 ng/mL solution in PBS was added to wells and incubated at 37 °C for 2 h. In control wells the RIP was added in the presence of 100 mM β-lactose. After thorough washing of the wells with PBST buffer, the conjugate of streptavidin with HRP diluted 10,000 times was added to the wells for 30 min at 37 °C. Then the wells were washed thoroughly, and TMB solution was added. The level of RIP binding to cells was estimated according to the absorption at 450 nm using SpectraMax i3x (Molecular Devices).

### 2.5. The Measurement of Ribosome Inactivation Degree by Ricin and Viscumin

The estimation of the ribosome inactivation activity of RIPs was performed as follows. The HT-29 cells were treated with a medium containing RIP in one of the following concentrations: 0, 1 × 10^−9^ M, 1 × 10^−8^ M, and 1 × 10^−7^ M for 1 h. Then, the cells were washed and the fresh medium without RIP was added and incubated further for 5 h. After that, the cells were washed with DPBS and lysed in Qiazol reagent.

The total RNA was isolated using a miRNeasy micro kit (Qiagen) according to the manufacturer’s instructions with DNase I treatment. The RNA integrity number (RIN) values for all samples were higher than 9.0. cDNA was synthesized from 500 ng total RNA per reaction using a SuperScript VILO cDNA Synthesis Kit. The qPCR was carried out using qPCR Mix HS Sybr (Evrogen), as described previously [25]. All experiments were carried out in three biological replicates.

RIP performs specific hydrolysis of the N-glycoside bond of adenosine 4324 of 28S rRNA, generating an AP site [18,26]. The reverse transcriptase incorporates a deoxyadenine opposite the AP site [27]. Thus, cDNA molecules synthesized from the modified and unmodified 28S rRNAs should differ by a single nucleotide (A instead of T), which can be detected by qPCR [27] (Figure 1).

Accumulation of the modified 28S rRNA was detected by monitoring the elevating signal of the modified cDNA fragment (MOD). Simultaneously with this, detection of the quantity change of the unmodified cDNA fragment (UNMOD) as well as of the total abundance of 28S rRNA (CTRL) was performed (Figure 1). The sequences of the primers used and the values of the qPCR efficiency of each primer set are presented in Table 1.

The fraction of modified 28S rRNA molecules was evaluated using the following algorithm of qPCR data processing. The value of the qPCR threshold cycle (*C_t_*) obtained for CTRL was used to normalize the *C_t_* values obtained for MOD and UNMOD in each RNA specimen. The fluorescence efficiency of the cDNA fragments MOD and UNMOD after SYBR Green staining should be the same since their nucleotide sequences are identical except for the strand symmetry of one base pair. Thus, the sum of their normalized *C_t_* values can be set at 100%. Then, the ratio of the normalized *C_t_* value of MOD to the sum of the normalized *C_t_* values of MOD and UNMOD was used to calculate the fraction of modified 28S rRNA molecules. Based on the assumption that a number of ribosomes is equal in each cell of the cell line, the obtained value of the fraction of modified 28S rRNA molecules was considered equivalent to the fraction of inactivated ribosomes in a cell.

### 2.6. All-Atom Molecular Dynamics

Spatial structures of the full-length ricin and viscumin and their catalytic A subunits (RTA and MLA) were extracted from PDB entries 5J57 [4] and 2RG9, respectively. Three N-terminal residues (1–3) in RTA and five C-terminal residues (249–253) in MLA were added using the Modeller 8.2 software [28]. MD simulations were performed in the GROMACS package [29] versions 5.1.4/2016.3/2018, compiled with CUDA GPU support, and all-atom AMBER99SB-ILDN force field. The TIP3P water model was used. The topology and force field parameters for chloroform and methanol were prepared previously [17]. MD simulations were carried out with a 2 fs time step and imposed 3D periodic boundary conditions in a dodecahedron box. Details of the simulation systems are given in Appendix A. The systems were equilibrated by the steepest descent energy minimization followed by heating from 5 K to 340 K or 310 K during a 1 ns MD run. Finally, long production MD runs (10 μs for RTA and 200 ns for the dimers) were carried out for each system. Electrostatic interactions were treated using the particle-mesh Ewald summation with fourth-order spline interpolation. The initial cutoff value of 1.2 nm and Ewald grid spacing of 0.12 nm were adjusted during calculations to balance the CPU-GPU loading. MD simulations were carried out in the isothermal-isobaric (NPT) ensemble with an isotropic pressure of 1 bar and a constant temperature of 310 K (RTA in water and dimers of ricin and viscumin) or 340 K (RTA in water, in a urea mixture, and in a chloroform–methanol mixture). The temperature and the pressure were controlled using the V-rescale thermostat and Parrinello–Rahman barostat with 0.5 and 1.0 ps relaxation parameters, respectively, and a compressibility of 4.5 × 10^−5^ bar−1 for the barostat. 

### 2.7. Monte Carlo Simulations with Implicit Membrane Model

MC simulations of several conformations of RTA—adapted to water (310 и 340 K), urea/water, and chloroform/methanol media—were carried out using the two-phase implicit solvation model [30]. The computational protocol was similar to that described in our previous study of MLA [17]. In the former case, the starting structure was that used in MD simulations in water (see above). For the urea and chloroform/methanol forms, the starting structures corresponded to one intermediate and the final configurations obtained via MD simulations in these solvents, as described above. The intermediate states were extracted from the corresponding MD trajectories after 5 µs.

Analysis of the RTA orientations with respect to the membrane was done using auxiliary programs specially written for this. The resulting MC states of RTA were analyzed using the following parameters: (i) total energy; and (ii) residue disposition with respect to the membrane (position (z) of its CA atom along the slab normal—axis *Z*). Residue i was considered interacting with the membrane if |z_i_| < 1.5 nm. 

### 2.8. Protein Surface Analysis

To characterize properties of the protein surface, dot Connolly surfaces of the catalytic A subunits were calculated with a probe radius of 0.14 nm (corresponding to water). In each point, a value of the molecular hydrophobicity potential (MHP) induced by protein atoms was calculated as described elsewhere [31]. The surface point i was considered hydrophobic if MHP_i_ > 0.3 (MHP values are given in logP units, where P is the octanol-water distribution coefficient). Analysis of the hydrophobic surface clusters was carried out using a grid-based clustering algorithm with a cell size of 0.3 × 0.3 × 0.3 nm^3^. Hydrophobic clusters were characterized by their surface area, composition (participating residues), and lifetime, as described previously [17].

## 3. Results

### 3.1. Comparison of the Cytotoxic Effect of Ricin and Viscumin on the Colorectal Adenocarcinoma HT29 Cell Line

In order to assess the cytotoxic effect of ricin and viscumin, the colorectal adenocarcinoma HT29 cell line was treated by one of the RIPs for only one hour and then the cells were cultured under standard conditions in RIP-free medium for five days. As a result, it was shown that the ricin cytotoxicity to HT29 cells was at least two orders of magnitude higher than to viscumin (the exact IC_50_ values were 1.3 × 10^−10^ M and 4.2 × 10^−8^ M, respectively; Figure 1a). The results obtained also indicated that in both cases a one-hour treatment of cells was sufficient for the penetration of such a quantity of RIP molecules that can cause cell death. However, the exact numbers of molecules needed to kill a cell appear to be different for ricin and viscumin.

### 3.2. Binding of Ricin and Viscumin to the Surface of HT29 Cells

It is generally accepted that binding of both ricin and viscumin to the cell surface occurs through specific interaction of the B subunit with glycoproteins and glycolipids exhibiting a terminal galactose residue [5,19]. However, differences in the binding specificity of the two lectins have been discussed [20]. To exclude the possibility that the difference in the toxicity of the RIPs on HT29 cells is due to different numbers of specific binding sites, we evaluated the binding of toxins to the HT29 cell surface using the In-Cell ELISA assay (Figure 1b). To ensure that binding of ricin and viscumin to HT29 cells is specific and occurs due to their lectin properties, as a control, RIP binding was investigated in the presence of lactose, which is a ligand for both RTB and MLB and prevents RIP binding to cells.

Note that the dissociation constants of ricin and viscumin binding to the cell surface are about the same [32]. Since the conditions of the cell-binding experiment were close to saturation, the obtained results suggest a conclusion that both RIPs show virtually equal numbers of the binding sites on the cells. Thus, the difference in their toxic effects on HT29 cells appears to be related to the difference in translocation efficiency across the ER membrane and the fate of the active toxins’ A subunits in the cytoplasm.

### 3.3. Comparison of the Ribosome Inactivation Capability of RTA and MLA

In order to compare the rate of RTA and MLA penetration into the cytosol, we estimated the fraction of inactivated ribosomes using the qPCR method after the treatment of HT29 cells by one of the RIPs for one hour and followed incubation for five hours in the medium without RIPs. The results averaged over three independent experiments are presented in Table 2.

According to the MTS results, a one-hour treatment is sufficient to cause HT29 cell death at all concentrations used for ricin (10^−9^ M, 10^−8^ M, and 10^−7^ M). However, in the case of viscumin, notable cell death was achieved only for the concentration of 10^−7^ M. The saturation of RIP binding sites on the cell surface was achieved at a concentration of 10^−8^ M [33]. Thus, the number of RIP molecules entering the cell might be significantly lower at 10^−9^ M in comparison with the other two concentrations. Since the enzymatic activities of RTA and MLA are similar [34] and the number of ribosomes is approximately the same in cells, the obtained results may indicate that the numbers of active toxin A subunits present in the cytoplasm per time unit are different for ricin and viscumin. The ratio of active RTA and MLA in the cytoplasm should be approximately similar to that for the fractions of inactivated ribosomes. Taking into account that RTA and MLA reach ER with similar efficiencies [35], we assume that the translocation across the ER membrane may be the step limiting the number of active catalytic subunits in the cytoplasm.

### 3.4. Structural Properties of RTA and MLA in the Disulfide-Linked AB-dimer

In order to be translocated across the ER membrane, RIP catalytic subunits have to refold. Therefore, we investigated the dynamics of this process for both toxins. However, before solving this problem, it was necessary to check how stable the A subunits were in the AB dimer in an aqueous environment. As has been mentioned above, both proteins have a high degree of structural homology (Appendix A). Proteins have similar helices (H1-H9 helices) and β-sheet regions (B1-B3 strands) with the exception of helix H2 (Figure 2a, encircled with a dotted line), which exists only in RTA.

During MD simulations of both disulfide-linked toxin heterodimers, the catalytic (A) and cell-binding (B) subunits were stable without significant perturbations of the secondary structure (Appendix A), with the root mean square deviation (RMSD) values calculated over heavy atoms with respect to the initial structure equaling 0.3 nm and 0.4 nm for RTA and MLA, respectively. Similar AB-dimerization interfaces of both toxins (Appendix A) were preserved in the course of MD and involved residues from H1, B2, H6, B3, H9, and C-terminal part of the A subunits (Figure 2b and Appendix A).

The distribution of the hydrophobic (Table 3) and polar properties (Figure 2c–e) of the RTA and MLA surfaces can serve as a pattern for their interactions with membranes or various chaperones involved in the transfer of the toxins through the membrane. In both proteins, a large non-polar cluster exists in the cavity formed by H7, B3, and H9 located at the AB-dimerization interface (Figure 2, cluster I). The second nonpolar region is located between the H1 helix and the N-terminal B1 strand (Figure 2, cluster II). In RTA, this region is partially immersed into the protein core, while in MLA it is significantly exposed to water. In RTA, the third large nonpolar zone is located between the termini of the H3, H4, and H5 helices and B1 strand (Figure 2, cluster III). In MLA, this region includes several small nonpolar clusters.

### 3.5. Conformational Behavior of RTA and MLA in Different Environments

In order to elucidate the molecular mechanism of penetration of the catalytic A subunits across the cell membrane, their conformational behavior in different environments mimicking the putative subsequent steps of toxin interaction with the lipid bilayer was simulated by MD. Figure 3 illustrates the evolution of the overall characteristics of the toxin molecules (RMSD from the starting model, gyration radius, and nonpolar surface area) during MD simulation in water, urea/water (equivalent of 8 M urea in water), and chloroform/methanol (1/1) mixtures. These solvents presumably mimic the toxin environments upon its transition from the bulk water and adsorption on the membrane surface followed by insertion into the membrane. In addition, structural rearrangements of the proteins in solvents having different polarities may occur during interactions with other proteins, which are involved in trafficking of RTA and MLA through the membrane.

Among the entire MD data set, both RTA and MLA demonstrated the highest conformational stability in water at 310 K—the corresponding RMSD from the starting structure did not exceed 0.3 nm, the radius of gyration and the fraction of nonpolar surface slightly fluctuated near their starting values, and the secondary structure of the toxin molecule was stable with some structural flexibility occurring only in the unordered C-terminal region (Figure 3 and Figure 4b; Appendix A). The total nonpolar surface and the size of hydrophobic clusters slightly increased (Table 3), as in this case they are not screened from water by the B-chain.

At 340 K, the flexibility of the interfacial part of both catalytic A subunits was slightly enlarged, accompanied by a small increase in variation of the macroscopic structural characteristics (Figure 3). The secondary structure of both toxins changed similarly (Figure 4c and Appendix A). The length of the H9 helix increased, whereas the H7 helix became unstable and transformed into a turn in RTA or into a π-helix in MLA. In addition, the conformation of the RTA B2 strand reversibly changed to “turn”. The flexibility of the C-terminal part of the proteins increased and spread further to the regions in contact with this fragment (B2, the loop between H8 and B3, Figure 4c), and some additional mobility appeared in the long loops of B1. The size of the hydrophobic clusters was similar to that observed in simulations at 310 K (Table 3). 

In the urea/water mixture, structural rearrangements of the A subunits occurred in a similar manner, but with a higher amplitude. In this case, the H7 helix completely melts in both toxins (Figure 4d and Appendix A). Moreover, destabilization of the B2 strand occurred both in RTA and in MLA. In RTA, a reversible unfolding of the B3 strand was observed, accompanied by fine-tuning of the structure of B3 and H7 (Figure 4d). In MLA, destabilization of this protein region was greater. After 2 μs, B3 separated from H7 and found another binding site on the protein surface (Figure 4d). This results in a complete loss of helicity for H7 (Appendix A), increasing mobility of the long loops of B1 (Figure 4d), and growing size of the hydrophobic cluster I1 (Table 3).

The largest structural rearrangements were observed in the chloroform/methanol mixture at 340 K (Figure 4e). Despite the relatively good stability of the secondary structure (Appendix A), after 10 μs the proteins completely lost their tertiary structure and turned inside-out. However, this process is much faster in RTA—it occurs within the first microsecond. To analyze the refolding pathway, we performed detailed analysis of the structural rearrangement in the toxin molecules during this period of time (0–1 μs of simulations) (Figure 5). In RTA, the structure transformation starts from its C-terminal part—similar to the picture observed under other studied conditions. In this case, H7 loses its contacts with H3 (first 400 ns), thus inducing a desorption of H3, H4, and H5 from B1 (400–800 ns) and fast transition of RTA into the molten globule state (after 1 μs). On the contrary, the contact of H7 and H3 is maintained in MLA during this period, and the observed structural transformation occurs much more slowly. As a result, the increased flexibility of B3 and H4 does not cause complete distortion of the MLA tertiary structure.

In order to verify that the observed behavior was not a modeling artifact, we carried out three additional 1 mks MD simulations of RTA and MLA in a chloroform/methanol mixture (see Appendix A). We found that although the order of movement of the protein parts depended on the simulation, their global behavior was reproduced. In all calculations, both proteins preserved their secondary structure. The RTA was more plastic than the MLA. In addition, we found that the loss of contact of the C-terminal part of the protein with the core observed for both proteins lead to core disruption only in the case of RTA.

Analysis of the surface shows that its non-polar fraction increased from ~5% to ~15% in both proteins by the end of the simulation (Figure 3c). Despite this, the character of the spatial distribution of such hydrophobic surface areas critically depends on the protein (Figure 6). In RTA, non-polar regions mainly form large clusters with a size greater than 6 nm^2^, while in MLA they are basically smaller than 2 nm^2^.

### 3.6. Interaction of RTA and MLA Adapted to Different Environments with Implicit Membrane: Monte Carlo Simulations

MD-conformations of RTA and MLA obtained from environments with different polarities presumably represent intermediate states of proteins during their transition from water to the membrane, up to the point of recognition by the ERAD machinery. However, in the MD simulations presented above, the membrane was not present—the calculations were performed in isotropic solvents. Therefore, the MD data alone are not sufficient to assess the ability of toxins to interact with the membrane and to identify potential sites of such binding. To solve these problems, we further tested the ability of RTA and MLA to interact with an implicit membrane model (the so-called “hydrophobic slab”) using Monte Carlo (MC) simulations. The latter were carried out in such a way that the spatial structure of the protein did not change significantly as compared with the starting one obtained either in water, in urea/water, or in chloroform/methanol mixtures. As a result, for each protein model adapted to a particular solvent, a large representative set of MC-states was obtained, and the lowest-energy ones were subjected to further analysis. The ability of a given protein model to interact with the membrane was evaluated in terms of the obtained patterns of its contacts with the hydrophobic slab, as shown in Figure 7. These details are given below.

#### 3.6.1. Water-Adapted Structures

As shown above, in the course of MD in water at 310 K and 340 K, RTA and MLA [17] do not differ significantly, so it was predictable that the MC results will be similar as well. It is seen that RTA prefers to stay in the polar environment, interacting with the hydrophobic slab by very few residues at the N-terminal (Figure 7a). The residues involved in the interaction do not belong to the described hydrophobic clusters, whereas the interaction of water-adapted MLA [17] with the membrane (Figure 7a) occurs through the N-terminal hydrophobic cluster II on the toxin surface. It is reasonable to conclude that the membrane binding of the water-adapted RTA via its N-terminal region is rather weak and therefore differs from that in the case of MLA [17].

#### 3.6.2. Urea/Water-Adapted Structures

The conformations of RTA and MLA [17] obtained after 5 and 10 μs MD runs in the urea/water mixture also undergo weak interactions with the membrane-mimicking hydrophobic slab (Figure 7b). Like the water-adapted RTA models, they have a single membrane-binding site, which includes the N-terminal residues. Nevertheless, this remarkably differs from MLA [17], where two membrane-binding modes were observed—due to partial unfolding of the helix H7 and the strand B7 adjacent to its C-terminus. The MLA [17] structure taken after 10-µs MD demonstrates much more prominent contacts with the hydrophobic slab (Figure 7b) via the residues 31–39, 46, and the C-terminal region 240–246, forming together a kind of two-site hydrophobic motif anchoring MLA in the membrane near the hydrophobic cluster I on the toxin surface.

#### 3.6.3. Chloroform/Methanol-Adapted Structures

MD simulations of RTA and MLA [17] in the chloroform/methanol mixture resulted in distinct global structural rearrangements accompanied by different exposure of multiple inner nonpolar regions of both proteins to the solvent. This allowed a much broader range of possibilities for interactions with the membrane (Figure 7c, Appendix A). After structural rearrangement in the chloroform/methanol mixture, the interaction of RTA and MLA with the membrane becomes energetically favorable. For both catalytic A subunits, a set of conformational states competing with each other was obtained, in which the toxin efficiently interacts with the membrane. Oppositely to RTA, which binds to the hydrophobic slab through multiple sites widely distributed over the toxin sequence, the membrane insertion of MLA [17] occurs mainly via its C-terminus, similarly to the case of the urea/water-adapted structure.

## 4. Discussion

In accordance with previous studies [18,35], we found ricin to have two orders of magnitude higher cytotoxicity than viscumin against HT29 cells (Figure 1a). This could not be attributed to differences between RTA and MLA enzymatic activities, which were insignificant [34]. Moreover, immunotoxins containing MLA are structurally similar to conjugates with RTA but have even higher cytotoxicity levels [21]. The discrepancy in the cytotoxicity level is often associated with a difference in the carbohydrate specificity of RTB and MLB subunits [20], which is supported by the different locations of ricin and viscumin on the cell membrane, suggesting that intracellular transport of these toxins starts from different membrane sites [35]. Thus, the number of RIP-binding sites on the cell surface appears to determine the rate of transportation into ER. Herein, it was shown that these numbers are close for ricin and viscumin (Figure 1b). Intracellular dissociation of the binding and catalytic subunits is also not the step determining the difference in cytotoxic activities of ricin and viscumin [36].

Therefore, it is reasonable to assume that the latter is due to specific conformational properties, namely, the plasticity of their catalytic subunits that affects the rate of translocation into the cytoplasm and proper refolding after crossing the ER membrane. Such an ability of RTA and MLA to adopt the initial globular structure to a changing environment (e.g., upon transition from water to the membrane interface) can presumably serve as a key factor regulating the toxin penetration into the cytoplasm. Indeed, before the translocation the structure of the RIP catalytic subunit undergoes some perturbations, and the protein globule gets partially or totally rearranged [10,37]. The hydrophobic C-terminal region that becomes exposed after a reduction in holotoxin in the ER has been shown to be important for the interaction of RTA with components of the ERAD system, and subsequent translocation of RTA to the cytosol [9,10,38]. Point mutation of hydrophobic Pro250 as well as addition of charged amino acid residues at the C-terminus of RTA diminished its transport from the ER to the cytosol [12,13]. Moreover, a temperature increase above 37 °C triggers a sharp increase in membrane binding accompanied by exposure of the protein C-terminus to the membrane [39] presumably induced by conformational rearrangements of the RTA. 

However, these data do not provide a consistent molecular picture of the structural rearrangements undergone by the A subunits of ricin and viscumin during translocation through the membrane. Moreover, they do not explain such a significant difference in the activity of RTA and MLA against ribosomes. In order to resolve this issue, structural/dynamic properties of both proteins should be evaluated and compared with atomistic details. Herein, this was achieved using computer modeling supported by the available experimental data serving as constrains of the modeling protocol. Thus, conformational behavior of both catalytic subunits was explored in their complexes with the B subunit and in the isolated form. In the latter case, an exhaustive comparative structural analysis of the proteins adaptation to different environments was also performed.

As a result, it was shown that in the AB-dimers the catalytic subunits are very stable in aqueous environments, which is consistent with experimental observations. In the absence of the B subunit, the surface of MLA is more hydrophobic than that of RTA (Figure 2d), which is consistent with our earlier biochemical experiments [21]. According to the present MD data, in polar environments (water and urea/water mixture) the globule of RTA is more stable compared to MLA. However, their exposure to the chloroform/methanol mixture (presumably mimicking the membrane interface, which combines both hydrophobic and polar environments) caused global structural rearrangements—the toxins turn inside out, although well preserving the secondary structure. By analogy with the terminology used in the field of protein folding, we will denote these states F (Folded) and U (Unfolded), respectively. (It should be borne in mind that in this study we do not consider the problem of protein folding; also, in our simulations the U-state simply represents the structure, which refolds in response to the solvent.) Therefore, the differences (if any) in conformational plasticity between the catalytic subunits of ricin and viscumin can only be related to their behavior in the membrane-mimicking medium approximated by the chloroform/methanol mixture. To explore this in detail, we compared the following parameters of the structural refolding (F → U) of RTA and MLA in this solvent: (1) the speed of degradation of the initial water-adapted globular state F towards the state U; (2) the “degree” of such a degradation; and (3) the hydrophobic/hydrophilic organization of the resulting U-structures. Based on the known principles of recognition of unfolded proteins by the ERAD components [40], it is reasonable to assume that in the case of RTA and MLA, this process is more efficient for a protein that unfolds faster (Criterion (1)), undergoes larger F → U conformational transition (Criterion (2)), and possesses hydrophobic properties, assuring its more efficient capture by the ERAD (Criterion (3)).

How are these conditions fulfilled for the resulting refolded models of both toxins obtained via MD in chloroform/methanol? Criterion (1): Upon contact with the membrane environment, MLA unfolds substantially slower than RTA (Figure 3b,c). This either directly affects the rate of its refolding and transportation or limits the spectrum of alternative components of the ERAD partners it can utilize in comparison with RTA. The latter is more susceptible to the structural rearrangements in hydrophobic conditions. A high degree of RTA plasticity is partially supported by experiments. Indeed, double mutation S215C/M255C causing S–S bonding, which prevents structural rearrangements exposing the toxin’s hydrophobic core, was shown to have an order of magnitude lower cytotoxicity, while maintaining the catalytic activity at the level of the wild-type toxin [10]. In order to compare properties of the proteins according to Criterion (2), several characteristics of the final refolded (U) models were used in order to find out which of them is more deviated from the initial globular (F) state. Among the analyzed parameters were the following: changes in the absolute and relative surface area (total, hydrophobic, hydrophilic) during the F → U transition; integral surface hydrophobicity parameters for the U-states (expressed in terms of surface MHP values); and the number of the residue-residue contacts lost upon the transition. It turned out that all these integral parameters were very close to each other for the U-states of RTA and MLA (data not shown), not allowing to discern which of the two proteins’ behavior is more consistent with Criterion (2). This seems rather strange, since in the presence of the implicit membrane the behavior of the U-state of RTA is strikingly different from that for MLA. As follows from the Monte Carlo simulations, RTA can bind the membrane by several pronounced hydrophobic sites distributed over the entire length of the toxin, while the U-model of MLA can only bind via its C-terminal part. In such a situation, a more detailed inspection of the U-states was performed—according to Criterion (3). As a result, it was clearly demonstrated that there were large differences in hydrophobic organization of these two protein models. Thus, despite the almost equal total area of the hydrophobic surface, the distributions of nonpolar regions are essentially different. In RTA, these regions mainly form large clusters, while in MLA the corresponding surface areas are represented by much smaller patterns (Figure 6). This explains higher ability of RTA to interact with the hydrophobic slab. 

To summarize, two main differences were found for the solvent-refolded states of the catalytic subunit of ricin as compared to viscumin: (i) a higher speed of the F → U rearrangement in the membrane-like environment; and (ii) a presence of large hydrophobic clusters on the protein surface, which are well suited for interaction with the membrane itself or with some other membrane proteins—putative ERAD components. Obviously, the latter is just a hypothesis. We therefore conclude that the efficiency of RTA translocation to the cytosol before ribosome inactivation appears to be greater than that for MLA. Taken together, the results of this study underlie the following putative scenarios of translocation of RTA and MLA across the ER-membrane to cytosol (Figure 8). The scenarios are based on the fact that direct translocation through the membrane and subsequent refolding without assistance of ERAD components is not enough to achieve the normally attained toxicity levels [9]. However, RTA and MLA partially refolded on the membrane surface can penetrate into ER and fold again with the aid of alternative ERAD retrotranslocation systems, more hydrophilic ERAD-L for proteins with a misfolded domain localized in the ER lumen, or lipophilic ERAD-M for proteins with a misfolded membrane domain [40] (see pathways shown by solid arrows in Figure 8b–d). Since the rates of ligand translocation by ERAD components increase with the degree of the ligand unfolding, the RTA with its higher susceptibility to refolding in hydrophobic conditions is better suited for utilizing ERAD machinery. Furthermore, higher plasticity of RTA would facilitate its refolding and activating the catalytic subunit after crossing the ER membrane, concurrently preventing its degradation, e.g., by avoiding the ubiquitination in the cytoplasm. Thus, translocation of the catalytic A subunits is likely to be the step limiting the toxicity of the RIPs. The developed multidisciplinary approach will be further tested on other RIP representatives both of type I and II for elucidation of the molecular mechanisms of their intracellular transport as well as for targeted rational design of new analogs of toxins with a therapeutic potential. 

## 5. Conclusions

Knowing the mechanisms of protein transfer to the cytosol is important both for understanding normal cell functioning and for developing novel approaches to targeted delivery of proteins and peptides. Penetration of water-soluble proteins into the cell is a complex multi-stage process dependent on many factors that require research using various methods. In this work, it was shown by biochemical methods that two homologous ribosome-inactivating proteins, ricin and viscumin, penetrate the cytosol with different efficiencies. A detailed study of these proteins using molecular modeling methods has revealed certain inherent features of ricin facilitating its passage through the membrane of the endoplasmic reticulum. Unlike viscumin, in the core of the ricin catalytic subunit there is an aromatic cluster that allows conformational rearrangements without loss of the secondary structure of the protein. More specifically, the higher cytotoxicity of ricin is determined (at least partly) by the plasticity and polar organization of its catalytic subunit undergoing partial unfolding upon interaction of the protein with the membrane surface, which is favorable for the ER-mediated translocation pathway into the cytoplasm. 

## Data Availability

Not applicable.

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
