# Peer review of "Differences in Medium-Induced Conformational Plasticity Presumably Underlie Different Cytotoxic Activity of Ricin and Viscumin"

_biomolecules, 2022, doi:10.3390/biom12020295_

Round 1

Reviewer 1 Report

This study investigates  conformational plasticity of  ricin and viscumin induced by the medium and its relevance to the cytotoxic activity of these toxins.

The introduction lacks a mechanistic description of the action of RTA - statement on the inhibition of ribosomes is not sufficient. Please kindly add the mechanism description (cleavage of nucleosides in the rRNA and maybe a figure describing this process would be very helpful). Maybe you may add this info to the Fig. 8.

Even though you provide lot of information in the discussion you must also supplement a clear Conclusion section - even though it is not obligatory, this will strongly improve comprehensiveness of your material.

References are not properly formatted, please consult instruction for authors and some recent paper in this Journal. All references must have also DOI supplemented (see the template).

Authors mostly cite Russian  references (16 from total 43 refs.)  - larger variability of the references must be provided. Referee cannot recommend respective references to be cited due to the ethical reasons, however, higher reference variability must be attained.

Typographic rules must be followed, e.g., latin names of species - l. 119-120 please correct - and also elsewhere. Material-methods - you must add also name of the towns (besides country) of the all suppliers. For liter - preferable L must be used. Between value and unit must be space (except of %).

Figure 1a - units at the axes must be given in [] parentheses.

Please, provide a list of abbreviations (preferably before references) - this will help reader to easily navigate in the paper.

The whole paper must be carefully doublechecked to remove numerous typos.

After major revision your paper can be reconsidered for acceptance.

Reviewer 2 Report

The authors investigate why ricin is more active than viscumin - two ribosome inactivating toxins, using biochemical and molecular modeling simulation strategies. These methodologies contribute to shed light on the structure-activity relationship of these toxins. Thus, this manuscript is interesting to readers dealing on several aspects of cytotoxins, particularly on the engineering of toxins for biomedical purposes. In addition, the manuscript brings practical examples of several programs of molecular dynamic simulation to interpret the interaction of the cytotoxin with biological membranes.

Before publication, it is recommendable a minor revision that includes:

1 - A revision of the English language, like spelling and semantics, is advisable.

2- How do the authors confirm that covalently linked biotin to the cytotoxins was 1:3 ratio?

3- Could the authors include the concentration of toxin as expressed in physical units (microgram per mL)

4- Specify what kind of HT-29 is? The origin of such a lineage.

5- Describe abbreviations the first time they appear in the text (e.g., RIP, PFA, HRP, TBM, AP site, ERAD, etc.)

6- Discussion about the qPCR and In-Cell ELISA results was missed and disconnected from the structural findings. In the discussion, it is recommendable to include the inference about the structure-activity relationship utilizing the experimental data described in this manuscript.

Round 2

Reviewer 1 Report

The authors have rectified all problems in the paper, which can be now accepted.